# Inequality can double the energy required to secure universal decent living

**Joel Millward-Hopkins** [1] ✉

Ecological breakdown and economic inequality are among the largest contemporary global challenges, and the issues are thoroughly entangled – as they have been throughout the history of civilisations. Yet, the global economy continues toward ecological crises, and inequalities remain far higher than citizens believe to be fair. Here, we explore the role of inequality, alongside traditional drivers of ecological impacts, in determining global energy requirements for providing universal decent living. We consider scenarios from *fair inequality* – where inequalities mirror public ideals – through a *fairly unequal* world, to one with a *super-rich* global elite. The energy-costs of inequality appear far more significant than population: even fair levels increase the energy required to provide universal decent living by 40%, and a super-rich global 1% could consume as much energy as would providing decent living to 1.7 billion. We finish by arguing that total population remains important nonetheless, but for reasons beyond ecological impacts.

The issues of ecological breakdown and economic inequality have become increasingly prominent in recent decades, forming the focus of multiple UN Sustainable Development Goals and social movements from Occupy Wall Street to Fridays for Future. If the warnings of researchers studying the evolution of civilisations are to be believed, this is a welcome development. Some researchers suggest that both ecological overshoot and extreme social inequality tend to accompany the collapse of civilisations[1]–as the material basis for meeting human needs is eroded, those at the bottom immediately feel the effects; in contrast, luxury consumption of elites continues until the bitter end, furthering overshoot[2].

Anxiety that civilisation is entering a period of severe decline isn't only found on the margins of academic discourse, and for the first time the scale is global. Concerns about global ecological breakdown are longstanding, but societies have yet to summon the capacity to substantially reduce anthropogenic impacts. We remain headed towards a climate emergency[3], while many other ecological impacts are at potentially catastrophic levels[4]. Researchers of Existential Risk have begun to take climate change seriously, highlighting the potential for climate impacts to cascade through food, provisioning, and political systems in ways that fragment the global cooperation required to address emissions[5] and heighten other existential risks (e.g. nuclear catastrophes)[6].

Inequality is thoroughly entangled with these dynamics. Climate change and other ecological impacts are driven by the Global North and affluent populations elsewhere[7,8], whose luxury consumption can be more difficult to decarbonise than that of those living at sufficiency[9]. And in highly unequal societies—where resentment and mistrust are widespread and the basic needs of many unmet—ecological shocks are more likely to translate into socioeconomic instabilities or violent conflicts[10–12]. The growing trend for global elites to purchase 'apocalypse real estate' in the apparent safe haven of New Zealand puts into sharp focus people's unequal capacities to adapt to ecological and socioeconomic instabilities (see, for example, Why Silicon Valley billionaires are prepping for the apocalypse in New Zealand; www.theguardian.com/news/2018/feb/15).

Clearly, interrelations between ecological breakdown and inequality are complex. However, the nexus between ecological limits, inequality and social stability remains understudied. Global ecological limits themselves are widely researched[13], and the implications for sustainable development of scenarios abiding by such limits are gaining attention[14–16]. Other research has explored current relationships between ecological impacts, inequality[17], and well-being[18], with some projecting relationships forward to assess how reducing inequality may effect ecological impacts[9,19,20] or to examine 'inequality corridors'[21]. Some have looked into the energy requirements of

[1]Sustainability Research Institute, School of Earth and Environment, University of Leeds, Leeds LS2 9JT, UK. ✉e-mail: joeltmh@gmail.com

providing good living standards universally[22,23]. Far removed from such research (with one exception[24]) are the social scientists investigating *inequality tolerance* and public notions of *fair inequality*[25]. But no studies have brought these fields together to consider scenarios where good living standards are provided to all within ecological limits, with inequalities low enough to ensure social stability.

Recently, we estimated that 40% of current global energy use would be sufficient to provide universal decent living standards in 2050 – standards assumed a prerequisite for high well-being[22]. But the Decent Living Energy (DLE) model considered a strictly egalitarian, high-population world with state-of-the-art technologies deployed everywhere.

Here the model is updated and extended to investigate the impacts of inequality. Various futures are explored, from one of *fair inequality* to a *fairly unequal world* with inequalities closer to those found today, alongside worlds of differing technological ambition and population growth. The notion of *fair* inequality considered herein acts as a reasonable minimum bound for inequality—higher equality would not find public support, while higher inequalities could generate popular resentment and social instability. To this end, the current work integrates public notions of fairness[24], alongside a bottom-up modelling approach that allows inequalities to be implemented directly in material terms (rather than indirectly via economic measures). The findings put into context the energy-costs of inequality alongside the traditional drivers of population and technology. Moreover, they show these costs to be considerable.

## Results

### Human well-being, decent living and energy use

A multiplicity of approaches have been used to explore the relationship between well-being and ecological impacts. One way these differ is in the concept of well-being employed. Some take broadly *hedonic* approaches, focusing upon happiness and subjective well-being (typically self-reported)[26]. Others take *eudemonic* approaches, considering multidimensional indicators of well-being informed by theories of human need[27]. Approaches vary further in their quantitative methods, which can be top-down or bottom-up. Top-down approaches use empirical data to investigate relationships between average national ecological impacts (e.g. energy use[28]) and social outcomes (life expectancy[29], Human Development Index[30]). A key finding is that while some countries manage to achieve good social outcomes with low ecological-impacts, none do so within planetary boundaries[18,31].

But an issue with top-down approaches is they inevitably start from observed relationships between social outcomes and ecological impacts, which emerge from current socio-political organisation with uneven global trade relations, high economic inequalities and the redundancies of consumer culture[32]. This limits their ability to explore transformative futures[33,34], where high social outcomes are secured universally with minimum impact. Bottom-up models can probe precisely such futures. They start from inventories of material consumption across key dimensions of human life – e.g. shelter, mobility, health – which together are assumed to meet human needs through providing *decent living standards*[35,36]. By estimating the energy-intensity of provisioning each aspect of these inventories, total energy requirements for a given population can be estimated.

Following pioneering work by Goldemberg et al.[37], recent research has estimated the energy required to secure decent living standards in key regions[23] and globally[22,38]. The current work updates this global model (see Methods), making static estimates of the global final energy requirements for providing decent living universally in 2050 (in line with our previous work) using the consumption inventory shown in Table 1, for the various scenarios described herein. These scenarios are not projections, but rather visioning exercises: they explore what's possible given technological developments on the one hand, and knowledge of basic human needs on the other.

## Modelling inequality (fair or otherwise)

An advantage of bottom-up models is that to study inequality, it must be explicitly incorporated. The DLE model included only need-based inequalities: inequalities in energy use were permitted only when population density, climate, or the age distribution of the population suggested that more (or less) energy use would be required to satisfy well-being. This work develops three less idealised scenarios (key features are summarised in Table 2) to compare with this strict equality:

The *fair inequality* scenario modifies the activity levels of Table 1 to mirror the *fair* levels of income inequality emerging from public value surveys[39], expanding upon an approach recently developed for carbon-footprint models[24] (see Methods). Crucially, the decent living standards provide a floor on consumption for the lowest consumers. Inequalities are then applied only to private luxuries (indicated by triangles in Table 1) and within countries. Between-countries, inequalities remain need-based—colder climates are permitted higher energy use for heating; more sparsely populated regions more mobility; but no countries were permitted higher activity levels or energy use simply for being richer. High-consumers aren't assumed to consume different goods, only more of what's in the inventory, making the results conservative. (It should also be noted that any notion of 'fair' inequality typically relies upon meritocratic values[25], which are easily contested[40]).

The *super-rich* scenario is identical to the *fair inequality* scenario, except for the top 1% in each country whose consumption of private luxuries is increased to that reported in a study of the carbon-footprints of the super-rich (see Supplementary Information). Essentially, this is a world of *fair* inequality for all but a small elite. Note, however, that the direct ecological impacts of the super-rich's lifestyles are a narrow conception of their influence, which plays out more fully through political power and investments in destructive industries[41,42].

Finally, in the *fairly large inequality* scenario, within-country inequalities in consumption are widened until they are closer to current levels for income inequality. Again, the lowest consumers are at decent living standards and between-country inequalities are only need-based, leaving global inequalities far lower than today; the global GINI coefficient of energy use in this scenario is 0.28, compared to 0.13 in the *fair inequality* scenario and the current level of 0.52[43]. Inequalities in this *fairly large* inequality scenario are somewhat arbitrarily defined (see methods for more details), but can be understood to be midway between what people deem fair and what currently exist.

## Population and technology

To compare the energy-costs of inequality with those of other drivers of consumption, scenarios of differing population and technological ambition are also developed:

For 2050 population, projections from the Shared Socioeconomic Pathways (SSPs) of the IPCC are used[44]. Putting aside massive catastrophes and global 1-child policies, the SSPs span the range of projections in the literature, which vary from ~8.4–10.1 billion depending upon compliance with educational & gender-equality related SDGs and how much the unmet-need for contraception remains unmet[45,46]. Here, the *high population* scenario uses SSP3 (10 billion in 2050) while for other scenarios SSP1 is used (8.5 billion in 2050). Estimates for global energy use scale almost linearly with population, but also vary with age composition and urbanisation.

For technology, the previous DLE model assumed universal provision of highly-energy-efficient, state-of-the-art technologies that are either currently available or will likely become so before 2050. The same assumptions are made for this work across except in the *current technology* scenario, where ambition is reduced so energy-efficiencies reflect current best-practice. For example, while the DLE scenario assumes highly efficient electric cars with a degree of automation for further efficiency benefits, the *current technology* scenario assumes

**Table 1 | Inventory of material consumption assumed to underpin the decent living standards of Rao and Min[35]**

| DLS dimension | Material requirements | Minimum activity levels | |
|---|---|---|---|
| Nutrition | Food<br>Cooking appliances<br>Cold Storage | 2000–2150 kilocalorie/cap/day<br>1 cooker/household<br>1 fridge-freezer/household | Δ |
| Shelter & living conditions | Sufficient housing space<br>Thermal comfort<br>Illumination | 15 meters² floor-space/cap*<br>Climate dependent<br>2500 lumen/house; 6 h/day | Δ<br><br>Δ |
| Hygiene | Water supply<br>Water heating<br>Waste management | 50 Litres/cap/day<br>20 Litres/cap/day<br>Provided to all households** | Δ<br><br>Δ |
| Clothing | Clothes<br>Washing facilities | 4 kg of new clothing/year<br>100 kg of washing/year | Δ |
| Healthcare | Hospitals | 200 meters² floor-space/bed | |
| Education | Schools | 10 meters² floor-space/pupil | |
| Communication & information | Phones Computers Networks + data centres | 1 phone/person over 10yrs old 1 laptop/household High** | Δ<br><br>Δ |
| Mobility | Vehicle production<br>Vehicle's propulsion<br>Transport infrastructure | Consistent with pkm travelled<br>4900-15,000 pkm/cap/year***<br>Consistent with pkm travelled | Δ<br>Δ<br>Δ |

Activity levels listed are those assumed for decent living standards (DLS) in the DLE model[22], but in the three inequality scenarios developed here, levels are varied for the categories indicated by the triangles (Δ) in the right-hand column (note that for food, kcal/day are fixed and GJ/kcal modified instead; see Supplementary Information).**
*Assuming 10 m² of living space/capita plus 20 m² of communal space per 4-person household.
***Activity levels here are not straightforward to define.
***Large range as this varies with regional population density.

**Table 2 | Summary of the key features of the six scenarios**

| Scenario | Population | Technology | Inequality | |
|---|---|---|---|---|
| | | | **International** | **National** |
| Decent living energy | 8.5 billion | State-of-the-art | Need-based only | Need-based only |
| High population | 10 billion | State-of-the-art | | Need-based only |
| Current technology | 8.5 billion | Current best-practice | | Need-based only |
| Fair inequality | 8.5 billion | State-of-the-art | | *Fair* levels based on public opinion |
| Super-rich | 8.5 billion | State-of-the-art | | *Fair* levels, except the top 1% |
| Fairly large inequality | 8.5 billion | State-of-the-art | | Commonly observed levels |

electric cars equivalent to the most efficient of those widely available now. This scenario thus remains highly ambitious, considering these technologies are deployed globally to all.

## Estimates of global final energy use

Global final energy use in the *decent living energy* scenario is 125 EJ (Fig. 1); ~70% lower than current levels. In the *high population* scenario this increases 18% to 148 EJ, due to the 16.7% increase in 2050 population from SSP1 to SSP3 and marginal ~1% increase in average per-capita energy use (largely due to less urbanisation). Energy use in the *current technology* scenario is higher still, increasing 47% to 183 EJ—a substantial change considering the conceptually small difference between this and the DLE scenario, but still under 50% of current global consumption.

Moving from the need-based inequalities of the DLE scenario to *fair* levels increases energy use by 40% (to 175 EJ); over twice the increase incurred by the *high population* scenario (Fig. 1). Adding a super-rich 1% of consumers increases energy use by a further 22 EJ, matching the increase of the *high population* scenario relative to DLE. This implies that the additional luxury consumption of the *super-rich* top 1%—above and beyond their consumption under *fair inequality*—uses as much energy as would providing decent living standards to the extra ~1.4 billion people in the *high population* scenario. Finally, energy use in the *fairly large inequality* scenario reaches 269 EJ; over double the DLE estimate, even though global inequality remains small

compared to today (due to the absence of significant inequalities between countries). Put another way, half of all energy used in the *fairly large inequality* scenario—and around one third in the *fair inequality* and *super-rich* scenarios—is for consumption beyond decent living standards.

Despite the significant energy costs of inequality, 2050 energy use in all scenarios remains far lower than many projections. For example, the most ambitious scenarios of the International Energy agency project 2050 final energy consumption of ~340–400 EJ, while some IPCC scenarios consistent with 1.5 °C warming approach 500 EJ[15] (Fig. 1b). However, this doesn't imply that the largest inequalities considered above are inevitably compatible with stringent (1.5 °C) mitigation, for two reasons. First, there are concerns about the feasibility of high-energy, 1.5 °C compliant pathways due to their heavy reliance upon unproven negative emissions technologies with substantial trade-offs[47]; systematic underestimations of rebound effects[48]; and hence broader questions about the ability to decouple ecological impacts from economic growth[34]. The scenario of Grubler et al. (2018)[15] offers a more realistic estimate of global final energy consumption consistent with 1.5 degrees, suggesting 245 EJ in 2050. Second, although even in the *fairly large inequality* scenario energy use is close to this level at 269 EJ, the technological assumptions remain highly ambitious and the population low. Moreover, when the *fairly large inequality*, *current technology* and *high population* scenarios are combined, 2050 energy use reaches ~460 EJ—close to the top of the IPCC range (Fig. 1b).

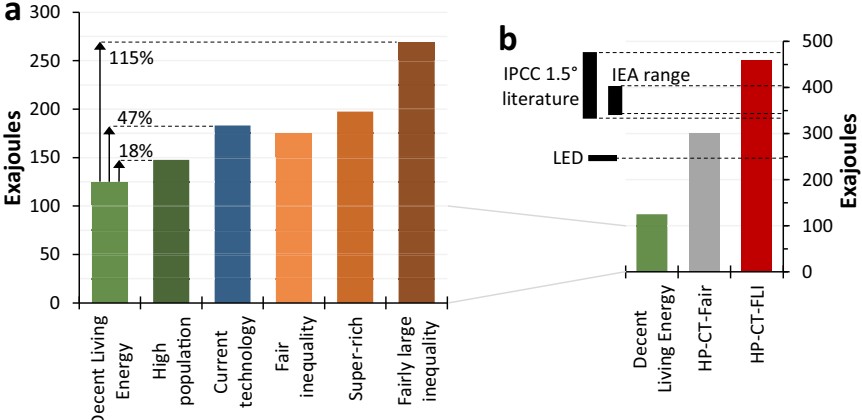

**Fig. 1 | Estimates of global final energy use compared to key scenarios from the literature.** Total final energy use in 2050 for the six scenarios (**a**). In panel **b**, the axis is expanded and three scenarios shown, including DLE and two combination scenarios: *high population, current technology* & *fair inequality* (HP-CT-Fair) and *high population, current technology,* & *fairly large inequality* (HP-CT-FLI). Various other scenarios are shown in b for comparison, including a range of 1.5°C consistent scenarios from IPCC literature[15]; the Net-zero by 2050 scenario (lower end of range) & Sustainable Development Scenario (upper end) of the IEA; and the Low Energy Demand (*LED*) scenario of Grubler et al. (2018)[15].

However, when the *fair inequality*, *current technology* and *high population* scenarios are combined, 2050 energy use is only 300 EJ—much closer to the Grubler et al. (2018) level.

## Energy inequality and composition

Globally, energy inequality in these scenarios is much lower than currently existing (Fig. 2). In the *fair inequality* scenario, energy use of the global top 1% (39.4 GJ/cap) is 2.7 times that of the bottom 10% (14.6 GJ/cap) who are at decent living standards. This ratio falls just within an inequality corridor suggested for Europe[21]. In the *super-rich* and *fairly large inequality* scenarios, the ratio climbs to ≈20 and ≈7, respectively, thus remaining considerably lower than the current ratio of ≈50[43]. In absolute terms, energy use of the top 1% in the *fair inequality* scenario remains below that of the bottom 20% in advanced economies such as Germany, Italy and Japan (*ibid*). Energy use of the top 1% in the *fairly large inequality* scenario (-105 GJ/cap) matches national averages in these same countries and is well below the current USA average. For the super-rich 1% (-300 GJ/cap) it's just below that of the current top 20% in the USA[43]. Finally, modelled energy use of those at decent living standards is similar to current averages in low-consuming countries of the Global South such as India, Tanzania and Ethiopia (15–17 GJ/cap). Of course, this does not imply decent living standards are being met here—indeed, the gaps are substantial[38].

Gini coefficients of energy consumption in the *fair inequality*, *super-rich* and *fairly large inequality* scenarios are also much lower than the current global level (>0.5[43]), but vary substantially across sectors (Fig. 3a). Shelter, water and mobility are where the largest energy inequalities are found—mobility-energy in the *super-rich* scenario is most unequally distributed—while nutrition-energy is most equally distributed in all scenarios. The overall sectoral composition of energy thus changes across scenarios, with the shares of nutrition and public services falling with increased inequality while those of shelter and mobility rise significantly (Fig. 3b). Specifically, in moving from the DLE to the *fairly large inequality* scenario the share of shelter in total energy use increases from 9% to 19%; in moving from the DLE to *super-rich* scenario the share of mobility increases from 17% to 26%. Recent empirically-grounded simulations also show a prominent shift in energy consumption towards mobility under increasing inequality[9]. Note that the present results follow directly from the assumptions made when implementing material inequalities, however, the assumptions are guided by literature—particularly saturation points for each category (see Supplementary Information).

## Regional energy use

Gaps between current energy footprints and our scenario estimates vary substantially across regions. Africa and India are the only regions where current footprints lie below even the DLE estimate for 2050 (Fig. 4a). In contrast, current footprints in the USA overshoot DLE by a factor of -10. In per capita terms, however, current footprints in Africa and India exceed DLE marginally, while the USA exceeds DLE by a factor of -13 (Fig. 4b).

Indeed, DLE per-capita is similar across regions, due to the assumption of only need-based intra-national inequalities (no notion of fair inequality between countries can be justified). However, the energy-costs of the three inequality scenarios vary with region, with slightly higher costs in Europe, the USA and China, where reported notions of fair inequality tend to be slightly higher. Note, however the lack of such data in African countries in particular, and the assumptions thus made (see Supplementary Information). Finally, the *current technology* scenario has similar energy-costs in each region as it does globally, while the energy-costs of SSP3 population growth vary predictably, being higher in regions where growth is concentrated (mostly India and Africa; Fig. 4a).

## Discussion

The key message emerging from this work is that inequality substantially increases the energy requirements of securing decent living standards for all. A world where decent living is secured universally, but material inequalities within countries remain close to current levels, could have twice the energy consumption of an egalitarian world with inequalities based only upon need. By comparison, moving from the lowest to highest plausible future population trajectory increases 2050 global energy use by only -18%. Consequently, if universal decent living is to be achieved and levels of inequality are not drastically reduced, meeting global climate mitigation ambitions will require either exceptionally ambitious technological deployment and low population growth, or substantial deployment of negative emissions technologies (or both).

An egalitarian world where material inequalities within and between countries are based only upon need is, of course, a fantasy. More realistically, inequalities could be reduced to levels broadly thought to be fair, and this would reduce the need for such high technological ambition and/or negative emissions. But current economic inequalities are an order of magnitude larger than people consider fair[39] (even without considering inequalities between countries).

And in the absence of drastic structural economic changes, they'll likely remain well beyond fair levels—partly because most are unaware of how unequal the societies they live in are[49]. Not only do these inequalities have severe implications for simultaneously meeting environmental and development goals – including climate and other planetary boundaries[50]—they also pose major justice concerns. For example, the energy consumption of a super-rich global 1% could equal that required to provide decent living standards to 1.7 billion people—the entire 2050 African population under SSP1.

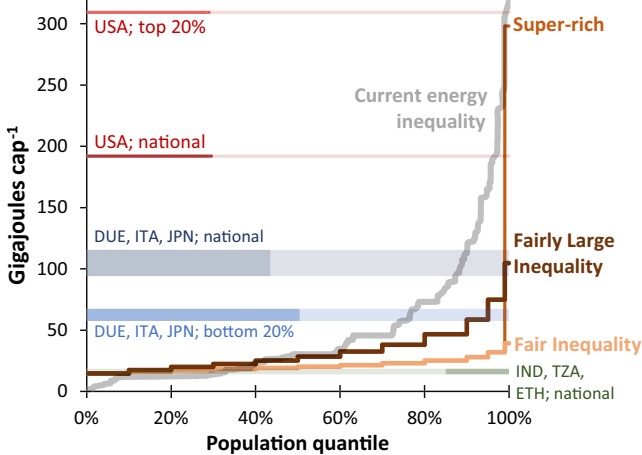

**Fig. 2 | Estimates of final energy use per capita across global population quantiles.** Final energy per capita in 2050 for the three inequality scenarios, shown averaged across population quantiles for the lowest to highest consumption groups. For comparison the current distribution of final energy is also shown, as well as that for a selection of income groups and countries (from Oswald et al. (2020)[43], but with government & capital energy added). The stepped patterns arise from the distributions produced in the inequality scenarios, which are at a resolution of deciles up to the top decile, which is split into three further groups (see Methods for more details). DUE Germany, ITA Italy, JPN Japan, IND India, TZA Tanzania, ETH Ethiopia.

Given all this, inequality should be considered as important as the traditional drivers of population, technology and (total average) affluence when analysing ecological impacts. However, the present work should not be used to argue population is unimportant, for three reasons:

First, large reductions in population growth would result from increasing women's bodily autonomy and, more broadly, achieving the SDGs[51]. Indeed, globally, half of pregnancies are unintended and in low-income regions 60% of these go on to become unplanned pregnancies[52]. But if conversations about population growth remain as heated as they've become, the political will required for extending family planning services may be diminished[53]. Second, those arguing overconsumption, not overpopulation, dominates ecological impacts reference the very low per-capita impacts in high-fertility regions today. But current consumption is irrelevant unless one condemns these populations to remain at their current living standards, which are incompatible with human flourishing. The important question asks what would be the ecological impacts of providing these people with good living standards, while reducing overconsumption of wealthy global populations. If living standards of the Global North and South converge (as they should), the impacts of an additional 3.5 billion in Africa and India by 2100—the difference between SSP1 & SSP3—become highly significant. Finally, however, this focus on impacts in relation to population growth distracts from the potentially larger issue of exposure to harm. Lower population growth in Africa, for example, may make little difference to global carbon emissions, but it would substantially reduce the population at risk of hunger in a warming world[54]. Moreover, while it's likely that lower population growth in poorer regions of the Global South won't slow the momentum of the global economy as it continues toward ecological crises[45], lower growth would mean there are far fewer people in the regions suffering the worst effects—and perhaps unable to escape, given the emergence of anti-immigration populist movements in the Global North that are driven, in large part, by discontent with current inequalities.

This work has probed the relationship between global ecological impacts, living standards, inequality, and public notions of fairness using a highly idealised model with many limitations. A concurrent advantage has been not being constrained by empirical data emerging from existing global political and economic structures, which have so

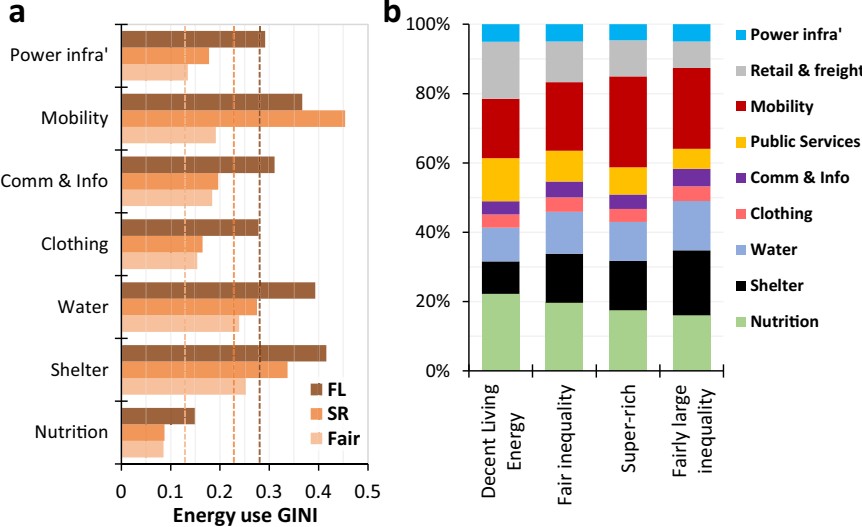

**Fig. 3 | Energy inequality for different sectors and the composition of total energy use.** Panel **a** shows GINI coefficients of 2050 global final energy use across consumption categories for the inequality scenarios (DLE is omitted as inequalities are negligible). The vertical lines indicate GINI coefficients for total energy use for each scenario. Panel **b** shows the sectoral breakdown of global energy use for the inequality scenarios, alongside the DLE scenario. Note, some of the legend is abbreviated: FL Fairly Large Inequality, SR Super-rich and Power infra' = Power infrastructure (i.e. the energy use involved in constructing power supply infrastructure).

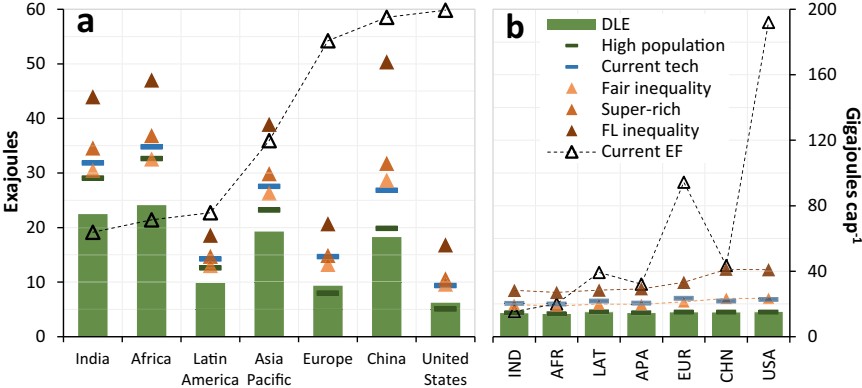

**Fig. 4 | Final energy use by region alongside current final energy footprints.** Final energy is shown in absolute (**a**) and per-capita terms (**b**). Current final energy footprints (EF) are from Oswald et al. (2020)[43]. Regions are ordered from lowest to highest absolute energy footprint, and dashed lines are only included where a trend is observed. Note that for visual clarity, the *super-rich* scenario is omitted from panel **b**; the data lies in between the *fair inequality* and *fairly large inequality* scenarios. Note also that the regions in both panels match, but names in **b** are abbreviated due to space constraints.

far proven unable to respond to the urgency of climate breakdown. However, this leaves enormous scope for further studies to explore both the real-world potential to reduce global ecological impacts by reducing inequalities, and practical means of realising such futures.

## Methods

### Modelling Decent Living Energy

The approach used to estimate global energy requirements is bottom-up, and involves combining activity levels for each material dimension of decent living with associated energy intensities, before summing across dimensions to obtain total final energy consumption. For example, for residential buildings, we have direct energy intensities for heating and cooling and indirect intensities of construction, all in MJ/m$^2$, which can be multiplied by the assumed activity-levels, in m$^2$/capita, to obtain energy use. The estimates thus include both direct energy use and the indirect energy required to produce products and infrastructures; the latter is divided by product/infrastructure lifetimes to give annualised values of indirect energy consumption. The calculations thus involve simple multiplication and summation. However, compiling the input data is an in-depth process.

The minimum activity-levels assumed in the scenarios are intended to describe what is appropriate for sufficiency; that required for decent living but no more. Rao and Min[35] offer the basis from which the values in the present work (and previous DLE model) are derived, and how these are translated to be suitable for an energy model is described fully in previous work[22]. Estimating energy intensities appropriate for state-of-the-art technologies as we intend requires harvesting and assimilating data from a broad range of sources including life cycle assessment, input–output analysis and industrial ecology. This process is also fully described in previous work[22]. For the present model, an analogous process is undertaken to obtain energy intensities for the *current technology* scenario, which is described in the Supplementary Information. Note also that both activity-levels and energy intensities are made regionally-variable where appropriate, and where suitable data exists.

The nature of bottom-up models is that some of the limitations of top-down models are avoided, but others are introduced. Most prominently, when one compiles an inventory of material consumption assumed sufficient for decent living, it is far more likely that important sectors will be missed than unnecessary sectors (mistakenly) included. The present model, for example, includes government services across healthcare and education, but it doesn't include police or military activity; it includes household consumption to support individual needs and social participation, but doesn't explicitly include artistic or cultural activities. It thus tends towards an underestimation of energy requirements. Other limitations relate to its static nature – the focus on

a single year – and consequent lack of assessment of the feasibility of state-of-the-art technologies being fully deployed by 2050, given the lifetimes and inertia of current infrastructures, not to mention social and political lock-in.

Finally, note that while global estimates of energy use are reported here, we only make estimates for 120 countries, which align with the GTAP regions used in the original DLE work[22]. However, these cover 89% of the 2050 global population and over 95% of current global GDP. And hence to obtain global estimates, we scale up the total energy use for the 120 countries by 112% (i.e. 1/0.89).

### Modelling final energy

Final energy is modelled as this better reflects the energy requirements of society and economic activity[55], as opposed to primary energy that captures losses during conversion of fossil fuels—e.g. coal into electricity, or oil into gasoline—losses that have no analogue for renewable energies. Final energy is, however, still a means to an end – specifically, to an energy service such as heating or mobility. These energy services provide benefits like comfort and social participation, which may satisfy various dimensions of human well-being. Decent living standards may thus not be broadly met even in societies with high final energy use (in excess of the DLE estimate) as energy use may be distributed highly unequally, provisioned via inefficient technologies, or directed towards energy services that are at odds with human well-being.

### Modelling 'fair' inequalities

Social and political scientists have studied public attitudes to inequality in various ways. We draw upon data reporting people's ideal income ratios between the highest earners and unskilled workers for 40 countries[39], of which 39 overlap with the 120 considered here (Iceland accounting for the difference). These ratios are lowest in Scandinavia and some Eastern European countries (at 2-3), higher in Germany and the USA (~7) and highest in Taiwan and South Korea (>10). Other important conclusions from this field of research are that notions of fair inequality are surprisingly consistent across countries, socioeconomic status, and political identities[39], and that almost all data suggests people significantly underestimate the extent of current inequalities[56].

We follow Millward-Hopkins and Oswald (2021)[24] to convert these maximum income ratios into idealised distributions considered to describe public notions of *fair inequality*. The first stage involves simplifying the approach by categorising countries as *egalitarians*, *moderates* or *meritocrats*, depending upon the level of inequality considered fair: *egalitarians* are countries where the reported ideal income ratio is under 4, *moderates* where it's 4-6, and *meritocrats* where it exceeds 6. We then produce idealised (lognormal)

distributions for each group, at a resolution of deciles up to the top decile, which is split into the 90–95ᵗʰ, 95–99ᵗʰ, and top 1%. Again following Millward-Hopkins and Oswald, for the three fair distributions, ratios between the top 1% and bottom 10% are set to 2.5, 5 and 8 for *egalitarians*, *moderates* and *meritocrats*, respectively, leading to the distributions shown in Supplementary Figure 1 (see Supplementary Information). For the *Fairly-large* inequality scenario, the countries remain categorised as either *egalitarians*, *moderates* and *meritocrats*, but the distribution for each county-type is widened until the GINI coefficients become 0.25, 0.35 and 0.45, respectively (up from 0.12, 0.21 & 0.26 in the *fair inequality* scenario). These values (0.25–0.45) are more in line with the range of national income GINI coefficients currently observed, although current values reach as high as 0.61 (in South Africa; see data.worldbank.org/indicator/SI.POV.GINI).

Sensitivity analysis shows our main results to hold even when these idealised distributions are parameterised differently, consistent with Millward-Hopkins and Oswald (2021). This leaves the major limitation being the amount of missing data–for 81/120 counties, largely in Africa and Asia–and we simply categorise these countries as *moderates*. Note, however, that the 39 countries for which we have data cover all six continents and include the world's major economies (e.g. China, the USA & most of Europe).

### Implementing inequalities in material consumption

In Millward-Hopkins and Oswald (2021), these idealised distributions were taken as income distributions, then translated into expenditure distributions and onto carbon and energy footprints using input-output data. For the present model, however, these distributions must instead be translated directly into material consumption.

To this end, the idealised distributions are taken as dimensionless descriptions of relative consumption. The distributions are thus applied linearly to private luxuries–housing size, car travel, air travel, hot water for bathing, energy-intensive foods, etc.–while retaining decent living standards as a floor on consumption for the lowest consumers. For housing, for example, the bottom 10% have 15 m²/cap of floor space, while the top 1% have 37.5 m² and 120 m² in *egalitarian* and *meritocratic* countries, respectively (i.e. $15 \times 2.5$ and $15 \times 8$). Mobility is more involved, as increases in private road transport are assumed to displace public surface transport before total mobility increases (see the Supplementary Information for details, and Supplementary Figure 2 for an example). To ensure this linear scaling didn't result in unrealistic values – there is only so many flights even the richest may take each year, for example–a sense check was done on the resulting consumption, and limits defined based upon the maximum expected even for the wealthiest (see Supplementary Table 3). For hot water, for example, a limit of 300 L/cap/day was applied, based upon flowrates of modern luxury showerheads. As mentioned above, for the *super-rich* scenario consumption of the top 1% was increased further for housing and mobility, to levels based upon those reported by Otto et al.[57] (also detailed in Supplementary Table 3).

The major assumption thus underpinning this process is that the income inequalities people believe to be fair can be taken to describe the inequalities in material consumption people think fair. Clearly this assumption can be challenged. However, for the present modelling approach–which is absent of monetary values–this is the only viable option, and it can be argued a reasonable approximation, as biases pull in both directions: On the one hand, applying inequalities only to a subset of the dimensions within the DLE consumption basket of Table 1, and not considering how wealthier classes will consume other luxury goods, biases the model towards underestimating the material inequalities that accompany income inequalities. Similarly, some of the things assumed to be equally distributed in the inequality scenarios are not so in reality – wealthier classes may draw more upon educational and healthcare services, for example, with children attending schools with smaller classes, and more frequent use of medical care,

directed not just towards health issues but also improvements (e.g. cosmetic surgery). On the other hand, however, income inequalities frequently manifest in ways that don't require additional material consumption – for example, expensive houses are of course not more expensive merely because they're larger, but due also to more exclusive locations. Such factors bias the present model towards over-estimates, as a proportion of income inequality will not manifest anywhere in the material consumption that the model considers.

## Data availability

The energy data generated in this study are provided in the Supplementary Information/Source Data file, and further data is available from the corresponding author on reasonable request. Source data are provided with this paper.

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

## Acknowledgements

This research was supported through a Leverhulme Trust Research Leadership Award granted to Julia Steinberger's *Living Well Within Limits* (*LiLi*) project (RL2016-048).

## Author contributions

J.M.H. conceived of the idea and analytical method, and undertook the modelling, analysis and preparation of the final manuscript.

## Competing interests
The author declares no competing interests.
