## [Peer Review File · Nature Communications]

Inequality can double the energy required to secure universal decent livingReviewers' Comments:

Reviewer #1:

Remarks to the Author:

The article is about providing a range of total final energy estimates of impacts of applying three selective energy consumption distributions in the future, where the full satisfaction of decent living standards (DLS) for the entire population is assumed. From what I know, this is the first work filling in the 'affluence' part of energy consumption estimate, in the world satisfying the decent living standards for all by adopting certain distributions, while other earlier works estimated just the requirements for minimum requirements for DLS. It is nicely presented, but the key message is a bit blurred. I see the main message is that this approximation still gives values that are still consistent with the level achieving 1.5C future and that even the future with 'fairly large' inequality can do so.

There are many other aspects presented, which makes it hard for readers to grasp the key idea the authors are trying to say. One important point presented was to juxtapose the role of population and the role of unequal distributions in determining total final energy use. Another was to emphasize the negative impact of the top-income population. And the role of the distribution that is perceived as fair as well as an update of decent living energy per capita etc. It gives an impression that the authors tried to patch many previous work components into one projection paper, which then damages the coherence of the story in return. If the key idea is to stress that the distribution matters for total energy consumption, I am sure it can be shown in a much more succinct way, e.g. even without bringing in a distribution perceived to be 'fair'. The work does not justify enough why such a distribution (somewhat arbitrary and hypothetic) is useful as a reference point for deriving the total final energy in 2050. (Or is it adopted as a sort of lower bound for the estimate? It will be helpful to express such a point.)

And I find the title is a bit confusing because the key results the paper is showing is not exactly "the energy requirements of providing universal decent living" but the requirements of meeting the affluence demand (i.e. beyond the decent living energy) especially when it is comparing the three inequality scenarios ('fair inequality', 'super-rich', and 'fairly large inequality') as shown in figure 2. For the three inequality scenarios, to do more justice to the original title, it should present how much of the total final energy is used for satisfying the decent living standards and how much for the living standards beyond that. And from my understanding, the former (which is truly the "energy requirements of providing universal decent living") is a constant across the three scenarios.

There are some minor clarification points below:

- If I understand correctly, the 'need-based' inequality within a country means everyone is consuming the same amount, since there are no need differences within a country. I am not sure whether the use of the term 'need-based' is correct in this sense intranationally. Or it is not clear enough in the text.
- I was not entirely sure why the last inequality scenario is named with 'fairly large', if it is reflecting the "current level of income inequality".
- Line 251-252: I wonder what the authors' definition of "affluence" is and how it is treated differently from "inequality" or "distribution" in the text.
- Line 270-273: "lower" pop growth [...] will leave far "more" people suffering the worst? Isn't it the other way?
- There is no remark on why 2050 is selected. An arbitrary year? What factors, other than the population, are dependent on time in this calculation?
- Line 111: referencing a wrong table
- Table 1: small triangles used without any explanations; "consistent with pkm traveled" is not informative enough to be used without any references.
- Figure 2: It can overlay the current distribution and the "decent living energy" scenario as reference points.
- Line 124-130 on 'fairly large inequality': I think this is the most 'realistic' scenario among the three, but it is very unclear what was done to make the distribution "closer to current levels for income

inequality". Does it mean it is stretched to achieve the same Gini as the national income Gini?

- Line 386: "some of the things assumed to be equally distributed in the inequality scenarios are not so in reality" <- It didn't say earlier what is assumed to be equally distributed.
- Line 196: "much lower than the current global level (>0.5)" <- there is no reference on the current global level.
- Line 175-178: A clarification is needed here. I guess this statement is meaning that those existing estimates from IEA or IPCC scenarios are based on explicit assumptions of these listed factors in one way or the other. And the authors' projections in this work are free from them because they are purely hypothetical values? It was not clear what the statement is trying to say.

Reviewer #2:

Remarks to the Author:

Thank you for giving me a chance to review the manuscript on "The requirements of providing universal decent living in a 'fairly' unequal world". The paper is well written and well structured. The methods are transparent. I have a few minor comments on the manuscript:

- Page 2 line 40-43: A reference supporting this statement is needed;
- The scenarios (e.g., Table 2) seem to be relatively similar, I wonder whether all scenarios are needed;
- I don't fully understand what "need-based only" in the column "International inequality" refers to. If the column is the same for all scenarios, is it necessary to include it in the table? Similarly, since the column on lowest consumers is the same for all scenarios, I am wondering whether it is necessary to include it in the table;
- How does international inequality interplay with national inequality? What are the assumptions on international inequalities? I think it is very important, large international inequalities will probably drive migration, and probably more inequalities within the migration receiving countries;
- Figure 1: the second figure panel is not entirely clear. Separation of the figure in Panel A and Panel B and a more clear explanation of Panel B would be helpful;
- Figure 3: It is not entirely clear why only three scenarios are represented in the left figure panel and all in the right figure panel. The figure caption is not clear. The shares in the right panel seem to be very similar. Perhaps presenting the absolute numbers instead of percentages would show better the differences in the scenarios?
- It would help to use the same colors or symbols for different scenarios throughout all figures. For example, the green color in Figure 4 is exactly the same green color that is used in Figure 3, however, it represents different parameters;
- The use of symbols and bars for different scenarios in Figure 4 is a bit confusing. For "Current EF" data points, rather than a curve should be used; Reducing the number of scenarios that are used in the manuscript could help to reduce the complexity of the figures;
- The discussion section could include some discussion which scenario is more likely and also which one would be more desired in terms of meeting the climate and sustainable development policy goals. Perhaps worth to connect in the discussion to the outcomes of the work of the Earth Commission. It is an international alliance of scientists asking questions about the environmental thresholds that the humanity should not cross in order to safeguard a planet that can support human well-being as well as how can be done in just way. Some outcomes of the Commission's work have been recently published, e.g.: Gupta et al. (2021) Reconciling safe planetary targets and planetary justice: Why should social scientists engage with planetary targets?, Earth System Governance, <https://doi.org/10.1016/j.esg.2021.100122>

Inequality can double the energy required to provide universal decent living

Response to Reviewers

Many thanks to both the editor and reviewers for taking the time to consider the manuscript. All of the feedback is detailed below in the left column, with my responses and details of revisions made in the right-hand column.

Reviewer #1

The article is about providing a range of total final energy estimates of impacts of applying three selective energy consumption distributions in the future, where the full satisfaction of decent living standards (DLS) for the entire population is assumed. From what I know, this is the first work filling in the ‘affluence’ part of energy consumption estimate, in the world satisfying the decent living standards for all by adopting certain distributions, while other earlier works estimated just the requirements for minimum requirements for DLS.

Thank you for the review of our work, and for the many constructive comments. I have addressed all of them and believe that they considerably improve the paper and in particular its key messages.

Major comments

It is nicely presented, but the key message is a bit blurred. I see the main message is that this approximation still gives values that are still consistent with the level achieving 1.5C future and that even the future with ‘fairly large’ inequality can do so.

Thank you to the reviewer for this comment. This was indeed how I had framed the results, but the message I aimed to convey is more nuanced.

Specifically, although the results show that even the most unequal scenario I consider is theoretically compatible with 1.5C, the energy costs of inequality are considerable (that is, if universal decent living is to be achieved). Consequently, if levels of inequality remain high, this will necessitate either extremely ambitious technological deployment and low population growth, or high dependence upon negative emissions technologies.

To capture this message, I’ve edited a key paragraph of the results (see page 9, paragraph 2) as well as the opening paragraph of the conclusions (page 13).

There are many other aspects presented, which makes it hard for readers to grasp the key idea the authors are trying to say. One important point presented was to juxtapose the role of population and the role of unequal distributions in determining total final energy use. Another was to emphasize the negative impact of the top-income population. And the role of the distribution that is perceived as fair as well as an update of decent living energy per capita etc. It gives an impression that the authors tried to patch many previous work components into one projection paper, which then damages the coherence of the story in return.

I have now reframed the results into what I hope is a far more coherent storyline (see page 9, paragraph 2; page 13, paragraphs 2 & 3):

1) Inequality substantially increases the energy requirements of universal decent living (more so than the widely accepted driver of population) and hence makes simultaneous achievement of climate and development goals far more difficult.

2) Reducing inequalities to *fair* levels – which represent a reasonable, realistic lower bound – would make global climate mitigation ambitions far less dependent upon extremely ambitious technological deployment and substantial exploitation of negative emissions.

If the key idea is to stress that the distribution matters for total energy consumption, I am sure it can be shown in a much more succinct way, e.g. even without bringing in a distribution perceived to be ‘fair’. The work does not justify

I hope this reframing captures the value of the *fair inequality* scenario. Further, the reviewer’s presumption is correct, namely that the *fair* inequalities considered act as a reasonable minimum bound for inequality – this reasoning is now elaborated upon in the introduction (page 3, final paragraph).

enough why such a distribution (somewhat arbitrary and hypothetical) is useful as a reference point for deriving the total final energy in 2050. (Or is it adopted as a sort of lower bound for the estimate? It will be helpful to express such a point.)	Personally, I also believe it is useful to draw explicitly upon public notions of fairness when studying inequality, given the contemporary polarisation of public discourse on political issues (i.e. the Culture Wars). Moreover, given that people from all sides of the political spectrum agree that current inequalities are too high, I believe it is useful to bring these considerations into energy and climate mitigation research (for further discussion, see https://iopscience.iop.org/article/10.1088/1748-9326/abe14f/meta). Finally, the calculation that equates the energy use of the top 1% with that of providing decent living standards to 1.7 billion is not intended to distract from the main story, but rather emphasise the importance of inequality and related justice concerns – this is now clarified in the discussion (page 13, paragraph 3).
And I find the title is a bit confusing because the key results the paper is showing is not exactly “the energy requirements of providing universal decent living” but the requirements of meeting the affluence demand (i.e. beyond the decent living energy) especially when it is comparing the three inequality scenarios (‘fair inequality’, ‘super-rich’, and ‘fairly large inequality’) as shown in figure 2. For the three inequality scenarios, to do more justice to the original title, it should present how much of the total final energy is used for satisfying the decent living standards and how much for the living standards beyond that. And from my understanding, the former (which is truly the “energy requirements of providing universal decent living”) is a constant across the three scenarios.	The reviewer is correct that the energy requirements of universal decent living is a constant across the three inequality scenarios. And because the first scenario (DLE) estimates this energy, the results the reviewer highlights are indeed presented, but only implicitly. Specifically, Figure 1 (left) shows energy use in the DLE scenario alongside the three inequality scenarios. The relative increases of the latter that I report are thus entirely associated with consumption beyond decent living. I have now reported the exact results the review suggests as well: for example, half of all energy used in the fairly large inequality scenario is for consumption above-and-beyond decent living (page 9, paragraph 1). This comment and the previous ones regarding framing have also made me reflect upon the title, and I’ve decided an earlier draft may indeed be more appropriate and more concise. I have thus changed the title to: Inequality can double the energy required to secure universal decent living
Minor comments	
If I understand correctly, the ‘need-based’ inequality within a country means everyone is consuming the same amount, since there are no need differences within a country. I am not sure whether the use of the term ‘need-based’ is correct in this sense intranationally. Or it is not clear enough in the text.	I agree that I had not defined need-based clearly enough. I have thus modified the description at two key points in the paper (page 5, paragraph 2; page 6, paragraph 1).
I was not entirely sure why the last inequality scenario is named with ‘fairly large’, if it is reflecting the “current level of income inequality”.	This is to some degree a play on words (fair inequality juxtaposed with fairly large inequality), but it is primarily because the scenario widens inequalities within countries to match current levels, but does not add any international inequality. This means global inequalities in this scenario aren’t as large as current global inequalities, thus the fairly large name appears appropriate.

Line 251-252: I wonder what the authors' definition of "affluence" is and how it is treated differently from "inequality" or "distribution" in the text	Here I am referring to total average affluence (i.e. the 'A' in the standard $I = PAT$ equation). I have clarified this in the discussion.
Line 270-273: "lower" pop growth [...] will leave far "more" people suffering the worst? Isn't it the other way?	Thank you for spotting this, it was a small but critical typo; I have now corrected it.
There is no remark on why 2050 is selected. An arbitrary year? What factors, other than the population, are dependent on time in this calculation?	This was chosen to be consistent with our previous DLE work in Global Environmental Change; that was, in turn, chosen as 2050 is often the end year in mitigation pathways such as the IEAs Net Zero by 2050. As the technological assumptions in the model are ambitious, looking ~30 years ahead also allows a feasible time window for their full deployment.
Line 111: referencing a wrong table	Now corrected
Table 1: small triangles used without any explanations; "consistent with pkm traveled" is not informative enough to be used without any references.	The triangles indicate the sectors for which material inequalities are implemented. This was previously mentioned in the table legend, but it was easily missed; I have now edited the description to make this clearer. It was also misreferenced in the main text, and this is now corrected as well (page 6, paragraph 1).
Figure 2: It can overlay the current distribution and the "decent living energy" scenario as reference points.	I have now added the current energy footprint distribution from Oswald et al. (2020). The DLE scenario is effectively there already, as inequalities are minimal such that it would be represented as an almost horizontal line beginning at the level of the inequality scenarios. Unfortunately there is not space to formally add this line, as it would overlay the current data for India, Tanzania & Ethiopia.
Line 124-130 on 'fairly large inequality': I think this is the most 'realistic' scenario among the three, but it is very unclear what was done to make the distribution "closer to current levels for income inequality". Does it mean it is stretched to achieve the same Gini as the national income Gini?	The reviewer is correct that the (lognormal) distributions are stretched until the GINI coefficients approximate current national levels – this is described in the methods and I have improved the description (page 17, bottom). An example for the UK is also shown in the supplementary information (section 1).
Line 386: "some of the things assumed to be equally distributed in the inequality scenarios are not so in reality" <- It didn't say earlier what is assumed to be equally distributed.	These are indicated in Table 1, but previously I'd not made this clear enough; see my comments above for corrections on this.
Line 196: "much lower than the current global level (>0.5)" <- there is no reference on the current global level.	This is from Oswald et al. (2020), who I also draw upon in the previous paragraph and in Figure 2. I've now added this reference to line 196 as well.

Reviewer #2

Thank you for giving me a chance to review the manuscript on “The requirements of providing universal decent living in a ‘fairly’ unequal world”. The paper is well written and well structured. The methods are transparent. I have a few minor comments on the manuscript:

Thank you for a positive review of the manuscript, and for the constructive feedback detailed below. I have revised the manuscript with all of these points in mind.

Minor comments	
Page 2 line 40-43: A reference supporting this statement is needed	A reference has now been added here.
The scenarios (e.g., Table 2) seem to be relatively similar, I wonder whether all scenarios are needed	I believe the scenarios each have their own specific value, and I assume it is only the inclusion of three separate inequality scenarios that the reviewer is questioning here so I will respond accordingly. Regarding the fair inequality scenario, I have now better justified its inclusion in response to comments from reviewer #1 (see page 3, final paragraph; page 9, paragraph 2; page 13, paragraphs 2 & 3). The fairly-large inequality & super-rich scenarios explore two different ways in which large global inequalities could manifest. I believe it useful to show both, as the latter makes justice concerns particularly salient, which I have now emphasised in the conclusions in response to reviewer #1 (see page 13, paragraph 3).
I don't fully understand what “need-based only” in the column “International inequality” refers to. If the column is the same for all scenarios, is it necessary to include it in the table? Similarly, since the column on lowest consumers is the same for all scenarios, I am wondering whether it is necessary to include it in the table	I have now elaborated upon the definition of need-based in response to comments from reviewer #1 (see page 5, paragraph 2; page 6, paragraph 1). As for Table 1, I believe it is useful to retain the international inequality column, even though the assumption is the same across scenarios, to avoid any misunderstandings regarding this aspect of the modelling. However, I have removed the lowest consumers column as suggested, given that this aspect of the modelling is clear from the main text.
How does international inequality interplay with national inequality? What are the assumptions on international inequalities? I think it is very important, large international inequalities will probably drive migration, and probably more inequalities within the migration receiving countries	Inequalities between countries are considered only to the degree that they reflect need, as in the previous DLE model. Now that the definition of need-based has been properly described, I hope this point is clearer. Indeed, the omission of international inequalities is the primary reason why even in the fairly large inequality scenario energy inequalities are lower than current levels. This is discussed at various point in the paper (page 6, paragraph 1; page 9, paragraph 1) and the point is clarified further by the addition (requested by reviewer #1) of current energy inequality to Figure 2.
Figure 1: the second figure panel is not entirely clear. Separation of the figure in Panel A and Panel B and a more clear explanation of Panel B would be helpful	I have increased the physical separation between these two figures, as well as changing one of the bars on the right figure (in response to reviewer #1) so there is less overlap between the information presented.

	I have also edited the legend so the figures are more clearly described.
Figure 3: It is not entirely clear why only three scenarios are represented in the left figure panel and all in the right figure panel. The figure caption is not clear. The shares in the right panel seem to be very similar. Perhaps presenting the absolute numbers instead of percentages would show better the differences in the scenarios?	The DLE case is omitted from the left fig. as inequalities are negligible, so reporting GINI coefficients would not be of value. I've added this explanation to the legend. As for the shares, I'm afraid I disagree that they are very similar. For example, moving from the DLE to fairly large inequality scenario the share of shelter in total energy use increases from 9% to 19%; from the DLE to super-rich scenario the share of mobility increases from 17% to 26%. I have now reported these numbers in the main text to highlight the key messages of the figure (see page 11).
It would help to use the same colors or symbols for different scenarios throughout all figures. For example, the green color in Figure 4 is exactly the same green color that is used in Figure 3, however, it represents different parameters	I have now modified colours in the sector plot of Figure 3 (right) so these don't match those used for the scenarios elsewhere. I've also modified the scenario colours in Figure 4 so that they match those in Figures 1-3.
The use of symbols and bars for different scenarios in Figure 4 is a bit confusing. For "Current EF" data points, rather than a curve should be used; Reducing the number of scenarios that are used in the manuscript could help to reduce the complexity of the figures	I have now edited this figure for visual clarity: the colours now match figures 1-3; the symbols have been made more consistent; and the super-rich scenario has been omitted from the right hand plot as the values were too cluttered and added little value here (a note on this has been added to the legend). I have retained the line for the current energy footprint data points as I believe it is useful to visually indicate this strong trend. However, I've made the line much less prominent (it's now thinner and dashed). See my response above regarding the reason for including all these scenarios.
The discussion section could include some discussion which scenario is more likely and also which one would be more desired in terms of meeting the climate and sustainable development policy goals. Perhaps worth to connect in the discussion to the outcomes of the work of the Earth Commission. It is an international alliance of scientists asking questions about the environmental thresholds that the humanity should not cross in order to safeguard a planet that can support human well-being as well as how can be done in just way. Some outcomes of the Commission's work have been recently published, e.g.: Gupta et al. (2021) Reconciling safe planetary targets and planetary justice: Why should social scientists engage with planetary targets?, Earth System Governance, https://doi.org/10.1016/j.esg.2021.100122	The discussion now mentions the (very low) likelihood of reaching the egalitarianism of the DLE scenario, the desirability of the fair inequality scenario, and the justice concerns inherent to the super-rich scenario given the hampered ability to simultaneously meet climate and development goals – here I now also reference the article the reviewer suggests. (See page 13, paragraph 3).